# Silicon-Based Superslippery/Superhydrophilic Striped Surface for Highly Efficient Fog Harvesting

**DOI:** 10.3390/ma16155423

**Published:** 2023-08-02

**Authors:** Xiang Ji, Shunxu Shuai, Shuai Liu, Yuyan Weng, Fengang Zheng

**Affiliations:** 1School of Physical Science and Technology, Jiangsu Key Laboratory of Thin Films, Soochow University, Suzhou 215006, China; 20204208008@stu.suda.edu.cn (X.J.);; 2SJTU-Pinghu Institute of Intelligent Optoelectronics, Jiaxing 314200, China

**Keywords:** superhydrophilic surface, superslippery surface, fog harvesting, striped pattern, heat transfer

## Abstract

Fog-harvesting performance is influenced by surface wettability, patterned structure and the heat transfer coefficient. In this work, we have prepared different surfaces with a stripe array of superhydrophilic, superslippery and superslippery/superhydrophilic surfaces for fog harvesting on silicon substrates using photolithography and silver-assisted chemical etching. The surface wettability and heat transfer coefficients of the above samples have been investigated. We analyzed the contact angle, sliding angle and transport state of droplets on these surfaces. The fog-harvesting rate of all samples under different voltages of the cooling pad (V = 0, 2.0, 2.5, 3.0, 3.5 V) was measured. Results showed that the superslippery/superhydrophilic striped surface could achieve rapid droplet nucleation, directional transport and efficient collection due to its superhydrophilic striated channels and the Laplace pressure difference between different wettability regions. At a condensation voltage of 3.5 V, the fog-harvesting rate efficiencies of the uniformly striped superhydrophilic and superslippery surface were 1351 mg·cm^−2^·h^−1^ and 1265 mg·cm^−2^·h^−1^, respectively, while the fog-harvesting rate of the superslippery/superhydrophilic striped surface was 1748 mg·cm^−2^·h^−1^. Compared with the original silicon surface, the maximum fog-harvesting rate of the superslippery/superhydrophilic striped surface was improved by 86.9%. This study offers significant insights into the impact of heat transfer and silicon surface wettability on the process of fog collection.

## 1. Introduction

As freshwater resources, such as rivers and lakes, become increasingly scarce due to pollution, it is particularly important to find another way to obtain freshwater resources that can be used by humans. As we all know, fog is a potential and abundant freshwater resource, and the use of fog-harvesting technology as a method of freshwater collection in arid areas has been proposed. In the past few decades, fog harvesting has received increasing attention in order to obtain clean water, and reports of fog-harvesting surfaces have been numerous [1,2,3,4,5,6,7,8]. The key factor influencing fog-harvesting performance is surface wettability, which is contingent upon the chemical composition and microstructure of the prepared surface. Consequently, researchers have developed numerous effective methodologies to engineer diverse surfaces, one of which involves emulating the microscopic structure of rice leaves. Rice leaves exhibit anisotropic wetting behavior, allowing for directional control of micro-water droplet movement due to their Cassie state [9]. In this state, droplets can freely roll with differing sliding angles along both parallel and perpendicular directions of the rice leaf. Taking inspiration from this natural phenomenon, scientists have fabricated arrays of stripes and grooves on various substrates to simulate the quasi-one-dimensional microstructure of rice leaves, enabling control over the directional transport of droplets. This approach has become a prevalent microstructural technique for investigating anisotropic wetting properties. For instance, Wu et al. employed photolithography, PDMS imprinting and micro/nanostructure coating methods to fabricate a surface combining micro/nanostructures and grooves. The micro/nanostructures enhanced the contact angle of droplets, thereby improving their migration rate, while the large grooves guided the droplets’ anisotropic sliding. They also investigated the impact of groove parameters on the sliding anisotropy in the Cassie state. The results demonstrated that larger groove cycles and heights led to more pronounced sliding anisotropy [10]. Besides the inherent structural parameters, different surface microstructures can also significantly affect fog water-collection performance. Seo et al. employed lithography and inductively coupled plasma (ICP) techniques to fabricate a surface with microscale columnar or gate structures on a silicon substrate. They then investigated the impact of the surface wettability of the microstructure on fog-harvesting performance. The results showed that, in the Cassie state, the pinning force of droplets on the groove array was smaller than that on the columnar array. Moreover, the average mass separating from the surface was smaller, and the falling frequency was higher, leading to significantly better water-collection performance on the groove array surface compared to the columnar surface [11]. However, during the process of fog harvesting, due to the high-humidity working environment, droplets are prone to transition from the Cassie state to the Wenzel state [9], resulting in a pinning effect that reduces fog-harvesting efficiency [12,13]. To address this issue, researchers took inspiration from Nepenthes and created a porous lubricating liquid-injected ultra-slippery surface. This lubricant significantly reduced the sliding resistance at the droplet–lubricant interface, thereby enhancing the migration rate of droplets during fog harvesting. Furthermore, due to the presence of the lubricant, droplets have difficulty penetrating the structure, thus impairing the surface performance. Superslippery surfaces have a wide range of applications, including fog harvesting, engineering and self-cleaning, etc. [14,15,16,17]. However, traditional SLIPS (slippery liquid-infused porous surface) exhibit isotropic droplet flow in all directions, and the movement direction of droplets cannot be controlled. By preparing micro/nanostructures and creating different wetting surfaces, the directional transport of droplets can be controlled [18,19,20,21,22,23,24,25]. For example, Dai et al. prepared a SLIPS with a gate-like nanotexture on a silicon surface using a modified black silicon method, and found that the smooth section can reduce droplet pinning, and the nanotexture can provide directional transport of water and thus improve fog-harvesting performance [26]. Biao et al. used a pulsed UV laser processing technique to fabricate a hydrophilic patterned SLIPS, featuring superhydrophilic stripes on a copper substrate, achieving highly efficient droplet transport and fog harvesting [27]. The above research works showed that the motion state of droplets can be effectively controlled by rationally designing and preparing hierarchical micro/nanostructures, so as to improve the efficiency of fog harvesting and droplet transportation. However, in these studies, the methods used by chemical or engineering technology are more complex. Therefore, the design of fresh-water collection devices with wide applicability, simple devices and low cost is an inevitable requirement for the further promotion of atmospheric freshwater collection technology.

In this study, inspired by rice leaves and Nepenthes, we employed a cheap and simple technique, a combination of a silver-assisted chemical etching method and photolithography, to fabricate a superslippery/superhydrophilic striped surface on a single-crystal silicon substrate, which achieved surface wettability differences and directional delivery of droplets without using modifiers, improving the efficiency of fog harvesting. The good thermal conductivity of single crystal silicon (~149 W m^−1^K^−1^) can further improve the fog-harvesting performance when collecting fog by condensation, and the dense silicon nanowire structure formed by etching can effectively fix the injected lubricant and prevent it from being lost during fog collection. In this work, we investigated the wettability and droplet motion behavior of different surfaces, such as anisotropic motion of droplets, instantaneous movement of droplets from superhydrophilic to superslipppery, sliding between two areas of different wettability, etc. The results showed that due to the synergistic effect of the Laplace pressure gradient and the superhydrophilic transport channels, the array of superslippery/superhydrophilic striped surfaces can effectively control the directional transport of droplets and improve greatly the efficiency of fog harvesting.

## 2. Materials and Fabrication

### 2.1. Materials

Silicon wafers (20 mm × 20 mm × 500 μm) were purchased from Lijing Technology Co., Ltd. (Taizhou, China). Hydrofluoric acid (GR, ≥40%), nitric acid (AR, 65–68%), acetone (AR, ≥99%) and hydroxide (AR, ≥99%) were purchased from Sinopharm Chemical Reagent Co., Ltd. (Shanghai, China). Hydrogen peroxide (AR, ≥30%) and anhydrous ethanol (AR, ≥99.7%) were purchased from Shanghai Lingfeng Chemical Reagent Co., Ltd. (Shanghai, China). Silver nitrate (99%) was purchased from Adamas (Shanghai, China). Photoresist (RZJ-304-25), Ag target (99%), and hydroxy-terminated polydimethylsiloxane (HTPDMS, viscosity of 25 cst at 25 °C) were purchased from Silicone Oil House Chemical Center (Shenzhen, China). All chemicals were used directly without further modification.

### 2.2. Solution Preparation

Solution A: Includes 15.6 mL of deionized water, 4 mL of hydrofluoric acid and 0.345 g of silver nitrate. Solution B: Includes 15.6 mL of deionized water, 4 mL of hydrofluoric acid and 1200 μL of hydrogen peroxide. Solution C: Includes 15.6 mL of deionized water, 4 mL of hydrofluoric acid and 0.0345 g of silver nitrate. Solution D: Includes 15.6 mL of deionized water, 4 mL of hydrofluoric acid and 400 μL of hydrogen peroxide. Solution E: including 15 mL of nitric acid, 7.5 mL of deionized water.

### 2.3. Fabrication of Homogeneous Striped Superhydrophilic and Superslippery Samples

Figure 1 shows the preparation process of four surfaces. The polished silicon wafers were cleaned with acetone, ethanol and deionized water, one after another, each for 5 min using ultrasonic cleaning. The photoresist was spin coated on the surface of the silicon wafer, and the mask of the stripe array was formed using photolithography technology. Next, the sample was immersed in solution A for 1 min to form a film of silver on the exposed area. Then, the sample was immediately taken out and immersed in solution B for 15 min to etch and form stripe array grooves. The width of the groove was determined by the width of the reticle pattern, and the depth of the groove was determined by the time the sample was immersed in solution B. After the etching was completed, the sample was immersed in solution D to wash off the silver film on the surface. The samples were taken out and ultrasonically cleaned with acetone to remove the photoresist on the surface. Then, the cleaned sample was put into solution C for 1 min, taken out immediately, and put into solution E for 15 min to etch the entire surface to form a nanowire structure. Solution D was used to wash away the silver particles on the surface, and the sample was then rinsed with deionized water. Finally, the sample was dried using nitrogen blowing to obtain a homogeneous superhydrophilic surface.

Based on this procedure, the HTPDMS lubricant was perfused onto the superhydrophilic surface, and any extra lubricant was eliminated by using spin coating to attain a superslippery surface.

### 2.4. Fabrication of Superslippery/Superhydrophilic Striped Surface

The process for preparing convex-superhydrophilic/concave-superslippery surface was illustrated in Figure 1b. First, a photoresist was reserved on the concave surface of the homogeneous stripe superhydrophilic surface during photolithography. Next, an Ag layer was deposited on the sample’s surface using the magnetron sputtering method. Here, a uniform and dense silver film was used to block the lubricant from contacting the superhydrophilic area of the exposed part. Then, the sample was washed with acetone to remove the photoresist and the silver film on the photoresist. HTPDMS was immediately poured onto the sample and then the barrier Ag layer was removed with dilute nitric acid, in order to expose the superhydrophilic region that was not perfused with HTPDMS. A stripe array sample with both superhydrophilic (convex) and superslippery (concave) was obtained. Figure 1c shows the preparing process for the concave-superhydrophilic/convex-superslippery surface, where the only difference was that the photoresist was reserved on the convex surface during photolithography.

### 2.5. Fog Harvest

Figure 2 shows the experimental installation for fog harvesting, which involves using a commercial ultrasonic humidifier to generate fog flow. A semiconductor cooling pad was affixed onto a bracket, onto which a sample was positioned vertically. The temperature of the sample was regulated by adjusting the voltage applied to the cooling pad. The surface temperature of the sample was determined using infrared thermography. The distance from the fog outlet to the sample was approximately 5 cm, with a fog flow velocity of approximately 3.4 m/s. Once the droplet reaches a specific size and weight, it detaches from the surface and falls due to gravity, ultimately being collected by a container positioned beneath the sample. The collected droplet is then weighed using a high-resolution analytical balance (METTER TOLEDO-AL204), with the difference between the container’s weight before and after collection representing the mass of the collected water droplet. The fog-harvesting rate, denoted as η, can be determined by the following equation:η = M/TS (1)

Here, M represents the mass of the harvested fog, T is the collection time and S is the surface area of the sample [28].

### 2.6. Characterization

The microscopic morphology of the surface and cross section was characterized and analyzed using scanning electron microscopy (SEM). Contact angles and sliding angles were measured using a contact angle meter. The motion of droplets on various surfaces was captured using a digital camera.

## 3. Results and Discussion

### 3.1. Morphological Features

Figure 3a displays an optical image of the superslippery/superhydrophilic striped surface. The light-colored area represents the concave, and the dark-colored area represents the convex. When the silicon wafer underwent a 1-min immersion in solution A, a continuous silver film was deposited on its surface. The surface was subsequently etched to create a micron-scale concave-convex stripe structure, as illustrated in Figure 3b. In solution C, the silicon wafer was immersed for 1 min, and this resulted in the formation of a vertical nanowire structure due to the discontinuous silver particles on the surface. Figure 3c,d show the cross section and surface morphology of the nanowire, respectively. The length of the nanowire is 7.32 µm. The red arrow in Figure 3d shows the measured nanowire diameter of 108 nm. The width of groove structure was determined by the mask pattern’s stripe width, and the nanowire length and concave groove depth were controlled by adjusting the etching time. The detailed results were depicted in Appendix A. The silver-assisted chemical etching’s detailed principle is presented in Appendix A.

To fabricate superhydrophilic patterns, we investigated the effect of nanowire length on the surface wettability of the etched regions (seen in Appendix A). When the sample was immersed in the etching solution for 2 min, the length of the nanowire was 1 µm, which was a superhydrophilic surface with a contact angle close to 0°. As the etching time was prolonged, the nanowire length also increased. We found that, as long as the length of nanowire reached 1 µm, it was enough to form a capillary phenomenon to prevent the loss of the lubricant. In this study the nanowire length of all samples was fixed at 7µm for better lubricant retention. We also characterized the samples infused with HTPDMS. As shown in Appendix A, HTPDMS was successfully injected into the nanowire structure, and OH hydrophilic groups were present on its surface.

### 3.2. Surface Wettability of Different Samples and the Behavior of Water Droplets

The sliding resistance of droplets moving on a surface is influenced by the surface’s wettability and patterned structure [27]. Here, we measured the contact angles of water droplets on various surfaces, including hydrophilic, superhydrophilic, superslippery and superslippery/superhydrophilic striped surface. As shown in Figure 4, the contact angle of a 5 μL water droplet on an untreated silicon wafer was approximately 68° in both parallel and perpendicular directions, obviously, which was no anisotropy. The surface of the untreated silicon wafer is naturally hydrophilic due to the presence of SiO_2_, making it smooth and homogeneous. Similarly, the contact angle of water droplets on the superhydrophilic surface is 0° in both parallel and perpendicular directions, indicating no anisotropy. Water droplets exhibited a film-like distribution on the superhydrophilic surface. When the striped superhydrophilic surface perfused by HTPDMS, the parallel and perpendicular contact angles of water droplets were 77° and 69°, respectively. The anisotropy of the contact angle was weaker on the superslippery surface due to a reduction in surface tension caused by the lubricant applied to the surface. The contact angle of droplets on the superslippery/superhydrophilic striped surface showed large anisotropy. On the convex-superhydrophilic/concave-superslippery surface, the parallel and perpendicular contact angles of water droplets were 82° and 33°, respectively. Similarly, on the concave-superhydrophilic/convex-superslippery surface, the parallel and perpendicular contact angles of water droplets were 77° and 36°, respectively. The droplets were distributed in an elongated shape on the 600 µm wide superhydrophilic stripes, as shown in the inset picture in Figure 4, indicating their tendency to spread along the direction of the stripes. However, the hydrophilic stripes generated energy potential barriers, limiting the droplets’ movement in the direction of the perpendicular stripes.

We investigated the sliding performance of both the superslippery surface and the superslippery/hydrophilic striped one with the contact angle anisotropy, as shown in Figure 5 and Appendix A. Figure 5a shows that a droplet took 433 ms to slide from the top to the bottom along the direction of the stripes on the superslippery surface when the tilt angle is 15°. In contrast, the same droplet took 8016 ms to slide in the direction perpendicular to the stripes, as shown in Figure 5b. This indicates that the droplet slip rate is higher in the parallel direction than in the perpendicular direction It can be explained by the fact that, on the superslippery surface, there is only the adhesion force of the lubricant and the sliding resistance generated by the stripe structure. The droplet needs only a small energy to overcome the sliding resistance moving in the perpendicular direction, so it can still slide slowly [28]. However, on the concave-superhydrophilic/convex-superslippery surface, the droplet movement time from the top to the bottom in the direction parallel to the stripes is only 368 ms, and at the same inclination angle, the droplet has a pinning effect in the perpendicular direction, as shown in Figure 5c,d, which indicated a very obvious sliding anisotropy. It can be attributed to the stripe interface between the superhydrophilic and the superslippery regions. When the droplet slides in the direction perpendicular to the stripes, it needs to overcome the significantly higher surface tension of the droplet-water interface [29]. These results suggest that the superslipppery surface is not favorable for controlling droplet motion, while the superslippery/superhydrophilic striped surface can better control the direction of droplet transport and further enhance the harvesting rate due to the existence of superhydrophilic stripes.

### 3.3. Fog Harvest

Fog harvesting includes three important processes, such as water capture, water transport and water removal. All the above processes are closely related to the wettability and patterned structure of the surface [30]. Furthermore, in the process of water capture, the rate of water capture is also dependent on the temperature difference between the fog environment and the sample surface. The fog-harvesting performance of the striped surfaces with superhydrophilic, superslippery and superslippery/superhydrophilic was recorded at different cooling voltages from 0 V to 3.5 V, as shown in Figure 6. The relationship between the semiconductor cooler temperature and cooling voltage can be seen in Appendix A. As a contrast with this, that of the hydrophilic sample without any stripes or post-treatment (raw Si wafer) was also shown here. In order to find the best arrangement of stripes, the effect of stripe width and concave depth on fog harvesting was studied (seen in Appendix A). Therefore, samples with a stripe width of 600 μm and a concave groove depth of 15 μm were selected for all tests. The samples were placed on a designed cooling plate in the laboratory, with a humidity of 70 ± 5% and a temperature of 24 ± 1 °C.

From Figure 6, we can find that the fog-harvesting rate of all samples increases gradually with the increasing cooling voltage (decreasing sample temperature), which is easily explained by the typical condensation theory [31]. It is noted that the fog-harvesting rate of the superslippery surface shows only a smaller increment than those of other samples. At a voltage of 0 V, the fog-harvesting rate of the superslippery surface is 1192 mg·cm^−2^·h^−1^, which is higher than that (850 mg·cm^−2^·h^−1^) of the superhydrophilic surface. However, at a voltage of 3.5 V, the fog-harvesting rate of the superslippery surface is 1265 mg·cm^−2^·h^−1^, which is lower than that (1351 mg·cm^−2^·h^−1^) of the superhydrophilic surface. It could be attributed, on the one hand, that the thermal conductivity of water film on the superhydrophilic surface, 0.59 W/(m·K), is greater than the one (0.14 W/(m·K)) of the lubricant on the superslippery surface. As shown in the Appendix A, under the same refrigerating voltage, the temperature of the superhydrophilic surface is lower than that of the superslippery surface, which results in a high fog-harvesting rate of the superhydrophilic surface. On the other hand, the lubrication ability of lubricants decreases largely with the decreasing of the temperature, which is manifested by the enlargement of both sliding angle and sliding time when the droplet is placed at the superslippery surface with 3.5 V cooling voltage, shown in Appendix A.

Obviously, at the voltage of both 0 V and 3.5 V, the fog-harvesting rate of the striped samples is enhanced largely, compared with those of the original silicon surface. The maximum fog-harvesting rate is found in the sample with striped concave-superhydrophilic/convex-superslippery surface, 1748 mg·cm^−2^·h^−1^ at the voltage of 3.5 V, which is 1.87 times of that (935 mg·cm^−2^·h^−1^) of the original silicon surface. The fog-harvesting rates of the striped convex-superhydrophilic/concave-superslippery surface are 1689 mg·cm^−2^·h^−1^ and 1603 mg·cm^−2^·h^−1^, respectively, at the voltage of 0 V and 3.5 V, both which are a little lower than those of striped concave-superhydrophilic/convex-superslippery surface, but higher largely than those of the striped superslippery or the superhydrophilic surface.

In order to better understand the fog harvesting performance, we used a high-resolution camera to record the process of droplets nucleation, growth, transport and drippage on the surface of all samples, as shown in Figure 7. Here, all samples were placed at room temperature. On the hydrophilic surface of raw Si wafer, as shown in Figure 7a, during the initial 30 s, some droplets undergo a slow nucleation process and then accumulate progressively. As a result, the volumes of more or less all droplets become larger and larger. After 45 s, the droplets with a critical size fall subsequently under the action of gravity, simultaneously, incorporating most of the smaller droplets along their sliding path. Finally, they accumulate together at the bottom of the sample and a large droplet grows to a sufficient size and then falls when the gravity of the droplet overcomes the edge tension of the sample. On the superhydrophilic surface, as shown in Figure 7b, when the small fog droplets accumulate continuously on the surface, there is no little water droplet, while a water film appears due to the film-like condensation. With the progression of time, a huge water droplet appears quickly at the bottom edge of the sample. Compared to the untreated hydrophilic surfaces, obviously, there are more nucleation sites on the superhydrophilic surface. Due to the existence of edge tension, however, the droplets are not easily shed (droplet pinning), which limits further improvement of the fog harvesting. It is interesting that the performance of ‘droplet pinning’ can be improved largely when the superhydrophilic surface is placed at about 5 °C, shown as in the Appendix A, which can be used to explain the fog-harvesting enhancement of the superhydrophilic surface, compared that of the superslippery sample with a cooling voltage of 3.5 V. Figure 7c shows the performance of fog harvesting of the superslippery sample at the room temperature. It is clear that the fog is condensed in the form of drops on the surface. After 30 s, due to the existence of stripes, the droplets will slide along the concave groove channel without growing to a very large diameter under the vertical direction. Obviously, a low sliding resistance of the superslippery sample is a major advantage. So, the superslippery surface can dislodge the condensed droplets faster than a hydrophilic surface, and, as a result, new water-free areas appear for further condensation. However, it is unfortunate that the sliding resistance of the superslippery sample increases at a temperature lower than room temperature; this is one of the reasons that the fog-harvesting rate of the superslippery surface only shows a smaller increment than those of other samples with the decrease of the cooling temperature.

In the absence of condensation, the nucleation rate of droplets, the size of the droplets leaving the surface, and the velocity of leaving the surface are the three major factors affecting the collection of fog [32]. According to Volmer’s nucleation theory, the nucleation rate of droplets on superhydrophilic surfaces is much higher than that on other surfaces [33,34]. However, due to the existence of nanostructures and the larger contact area of the surface, the formed droplet also has a larger contact angle hysteresis and a larger critical mass when the droplet leaves the surface, which may offset its faster nucleation rate, as confirmed in Figure 7b. Our fabricated striped superslippery surface, the droplet size is limited by the presence of grooves, and the hydrophilic lubricant increases the droplet migration rate. As shown in Figure 7c, compared with the striped superhydrophilic surface, the droplets hanging on the bottom of the sample are smaller and leave the surface faster, so the fog harvesting efficiency is greater than that of the superhydrophilic surface. As shown in Figure 7d,e, at 15 s, it was observed that there were more droplets on the concave superhydrophilic/convex superslippery surface compared to the bottom of the convex superhydrophilic/concave superslippery surface. This is because the superhydrophilic region is more conducive to the transport of droplets than the convex superhydrophilic region. Compared with the striped superhydrophilic surface, although the size of the droplet leaving the surface is almost unchanged, the existence of the superhydrophilic stripes creates a curvature gradient of the liquid film, so that the droplet is faster along the hydrophilic groove for effective capillary pumping [32], so as to achieve better fog-harvesting efficiency. In summary, the superhydrophilic/superslippery striped surface is more conducive to the rapid collection and departure of droplets from the surface, thereby improving the fog-harvesting rate. Note that droplets nucleation only happen on the concave-superslippery stripes or convex-superslippery ones, indicated by the red ellipse in Figure 7d,e, and no any droplets appear on the concave-superhydrophilic stripes or con-vex-superhydrophilic ones.

To further comprehend the movement of the droplets on superslippery/superhydrophilic striped surface and superslippery one during fog harvesting, a series images of droplets were captured during the process of the appearance, growth, merging with each other and disappearance (seen in Appendix A). In Figure 8a, during the fog collection process, small liquid droplets appear on the convex surface of the superslippery striped sample and then transfer to the concave grooves. Once a droplet grows, accompanied usually by the merging with each other, to a size constrained by the stripes, it is inclined to slide down along the superslippery concave grooves until it leaves the surface along the concave grooves. This whole process takes about 17,760 ms.

As shown in Figure 8b, on the concave-superhydrophilic/convex-superslippery surface, a dropwise condensation happens in the concave superslippery regions and a filmwise condensation does in the concave superhydrophilic ones. As the little droplets grow, they move randomly in the superslippery regions and combine with each other. When a combined droplet touches the superhydrophilic region, it quickly moves to the superhydrophilic stripe under the action of the Laplace pressure difference [26]. This whole process takes about only 3090 ms, far less than that (17,760 ms) of the superslippery surface. Similarly, on the concave-superhydrophilic/convex-superslippery surface (not shown here), this droplet transfer also happens from the superslippery regions to the superhydrophilic. Note that, the above-mentioned droplet transfer can happen when the droplet has only a small size, as a result, a new water-free areas appears on the superslippery regions, which inevitably facilitates further droplet nucleation and leads to quick water delivery to the bottom of the sample [35]. Overall, droplet nucleation, growth and transport are significantly improved due to the presence of superhydrophilic stripes between the superslippery stripes, which results in a better fog-collection performance than that of the superslippery surface or the superhydrophilic one.

## 4. Conclusions

In summary, the superslippery/superhydrophilic striped surface was prepared on silicon substrate by combining lithography technique and simple silver-assisted chemical etching method. The motion state of droplets is studied on superhydrophilic surface, superslippery surface and super hydrophilic/superslippery surface with stripe array in experiments. According to the findings of the study, the fog-harvesting rate on the superslippery surface of the striped array increases gradually with the increase of cooling voltage. However, at a cooling voltage of 3.0 V, the fog-harvesting rate on this surface begins to be lower than that of the superhydrophilic surface. It is due to the fact that the heat transfer coefficient of lubricant is smaller than that of water, and the sliding resistance of lubricant increases with the decrease of temperature. Note that the contact angle and sliding angle of the superslippery/superhydrophilic striped surface have the best anisotropy, which is ascribed to the larger surface tension of the superhydrophilic stripes and the lower sliding resistance of the superslippery stripes. The fog-harvesting rate was highest on the superslippery/superhydrophilic striped surface, which is due to the fact that the droplets in the superslip region can quickly transfer to the superhydrophilic region and slide down. This study offers significant insights into the impact of heat transfer and silicon surface wettability on the process of fog collection.

## Figures and Tables

**Figure 1 materials-16-05423-f001:**
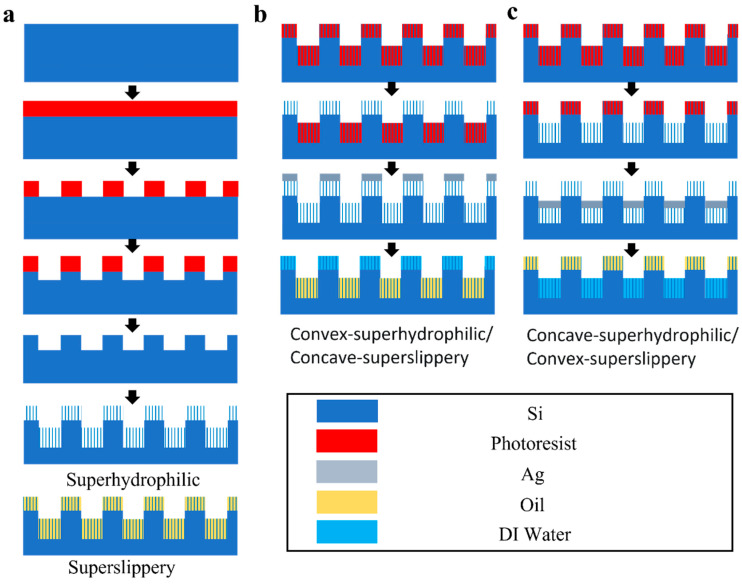
Flow chart for preparation of different surfaces. (**a**) Striped Superhydrophilic and Superslippery surfaces (**b**) Striped convex-superhydrophilic/concave-superslippery surface (**c**) Striped concave-superhydrophilic/convex-superslippery surface.

**Figure 2 materials-16-05423-f002:**
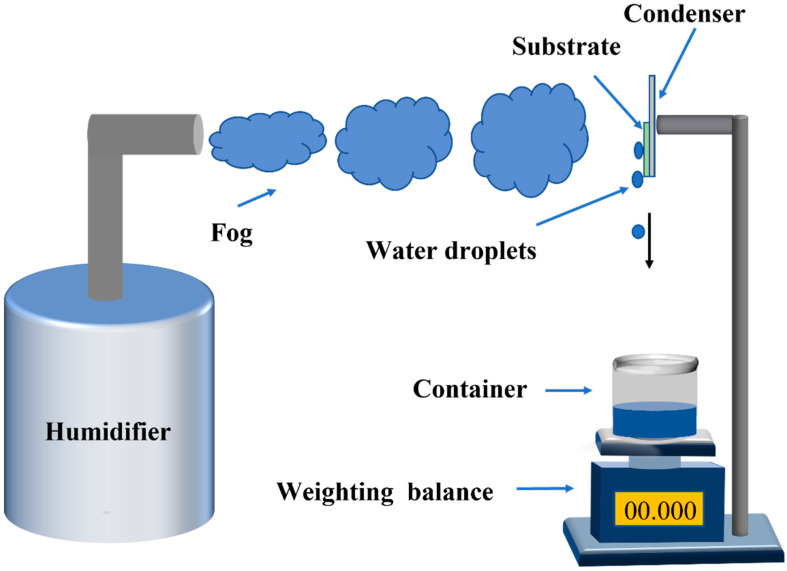
Schematic diagram of the fog harvesting device.

**Figure 3 materials-16-05423-f003:**
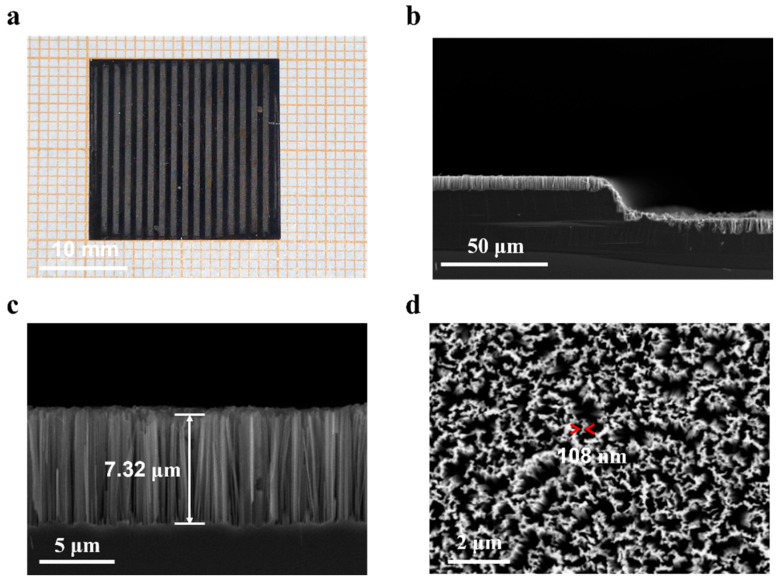
(**a**) Optical image (**b**,**c**) SEM cross section image (**d**) SEM surface image of super hydrophilic region before filling lubricant.

**Figure 4 materials-16-05423-f004:**
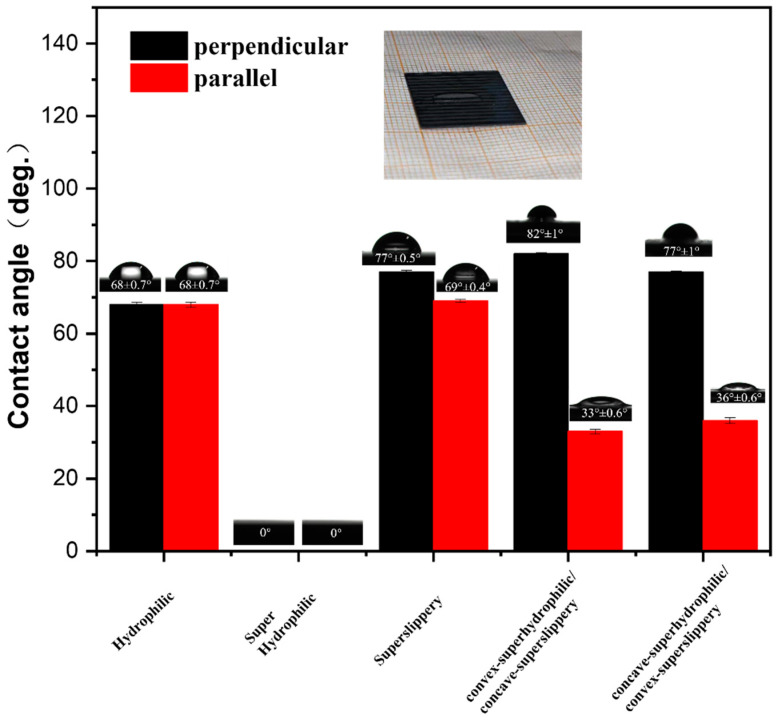
Contact angles of a droplet (5 μL) in parallel and perpendicular directions on hydrophilic, superhydrophilic, superslippery, superslippery/superhydrophilic striped surface.

**Figure 5 materials-16-05423-f005:**
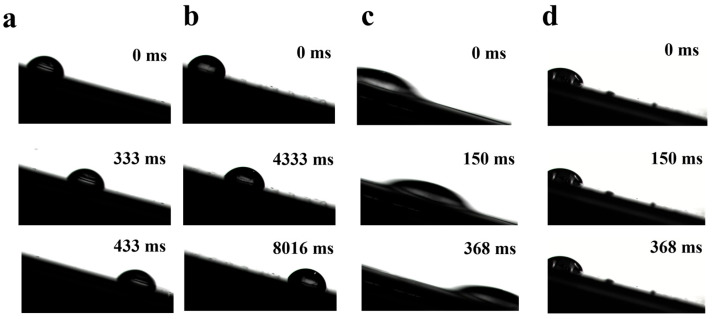
Time-lapse images of a 5 μL droplet on a surface at an inclination angle of 15°. The direction (**a**) parallel and (**b**) perpendicular, respectively, to the stripes on the hydrophilic superslip surface. The direction (**c**) parallel and (**d**) perpendicular, respectively, to the stripes on the concave-superhydrophilic/convex-superslippery surface.

**Figure 6 materials-16-05423-f006:**
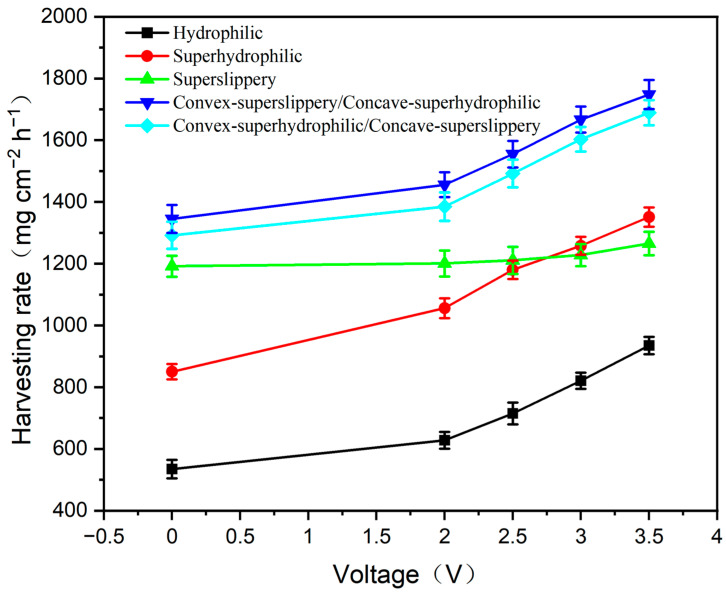
Fog-harvesting rate of five samples. The back surfaces of all samples were pasted with a semiconductor cooler, on which different voltages were applied to control the temperature of the sample’s front surface toward the fog.

**Figure 7 materials-16-05423-f007:**
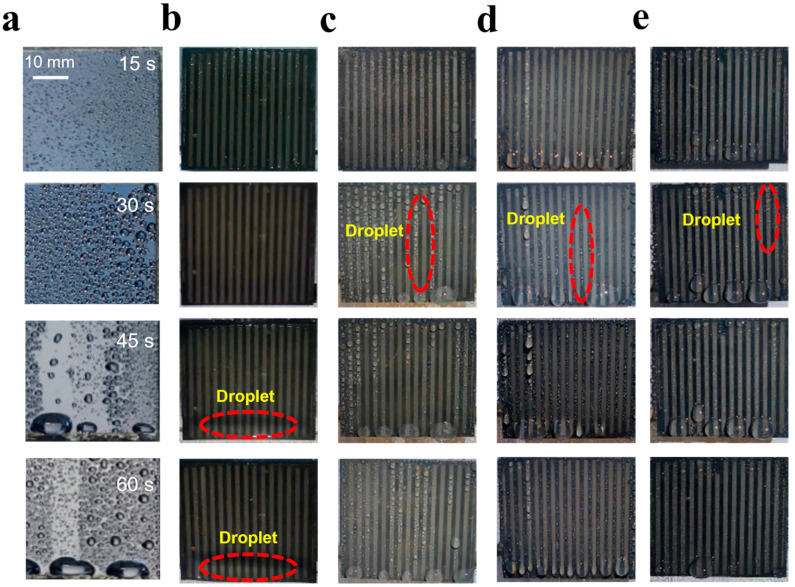
The movement state of droplets during fog harvesting (**a**) hydrophilic surface (**b**) superhydrophilic surface (**c**) superslippery surface (**d**) convex-superhydrophilic/concave-superslippery surface (**e**) concave-superhydrophilic/convex-superslippery surface (during the test, all samples were placed vertically).

**Figure 8 materials-16-05423-f008:**
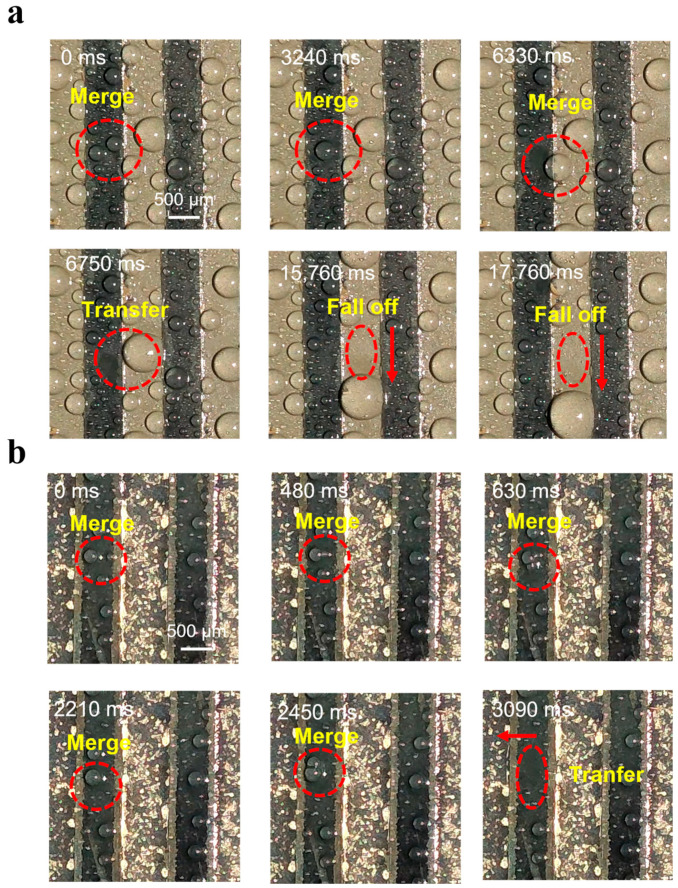
Moving image of the condensation droplets on (**a**) superslippery and (**b**) concave-superhydrophilic/convex-superslippery surface with a cooling voltage of 3.5 V when fog is collected.

## Data Availability

The data that support the findings of this study are available from the corresponding author upon reasonable request.

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
