# Peer review of "Silicon-Based Superslippery/Superhydrophilic Striped Surface for Highly Efficient Fog Harvesting"

_materials, 2023, doi:10.3390/ma16155423_

Round 1
Reviewer 1 Report
The manuscript presents an experimental investigation of surface wettability on patterned silicon surfaces, resulting in superhydrophilic, superslippery and superslippery/superhydrophilic behaviors for fog harvesting applications.
The manuscript is well written, the experimental data clearly presented and the results scientifically sound and interesting.
I believe the topic is well suited for the audience of Materials and I therefore recommend the manuscript for publication.
I would like however to encourage the authors to better clarify in the introduction why they chose the stripe pattern design for such applications, preferring this with respect to other pattern shapes. Also, how do the pattern parameters and geometry affect the results?
Reviewer 2 Report

Minor language correction is required.
Reviewer 3 Report
The authors have put in a commendable effort in preparing the manuscript, which is sound and of interest to the scientific community. However, I have a few concerns that need to be addressed before the manuscript can be published.
Firstly, although the results achieved using super-slippery/super-hydrophilic striped surfaces were superior, the literature review lacks clarity in explaining the characteristics and recent efforts to enhance performance. Additionally, the novelty of the structure in comparison to recent studies is not clearly highlighted.
Secondly, while the results are adequately depicted, there are instances where the scientific/physical phenomena behind them require further explanation. For example, in line 335, the phrase 'we think that' should be supported by scientific reasoning and justified adequately, rather than being based solely on personal opinion.
Furthermore, in Figure 6, the harvesting rate for all surfaces, except the super-slippery surface (the green curve), remains constant without any deflection at 2Vs. It would be beneficial to provide a clear explanation for this behaviour.
There are also a few minor issues that should be addressed:
- Please verify if the 'S' in Fig. S4 and S5 is a typographical error or provide an explanation (Lines 257-258).
- The marker in the black curve of Figure 6 is missing an error bar.
- The in-text citations are confusing. The placement of the full stop before the reference is incorrect; it should follow the reference.
- The references should be rewritten according to the journal's standard. There is no need to indicate the type of article (e.g., [j]) in the references.
Lastly, please correct the following English language errors/typos:
- In lines 284-285, 'little a lower' should be 'a little lower'.
- In line 293, 'to understand better' should be 'to better understand'.
Quality of English, although understandable needs a thorough revision.
